# Renal Tubular Glucagon-Like Peptide-1 Receptor Expression Is Increased in Early Sepsis but Reduced in Chronic Kidney Disease and Sepsis-Induced Kidney Injury

**DOI:** 10.3390/ijms20236024

**Published:** 2019-11-29

**Authors:** Jae Hyun Choi, Seung Jung Kim, Soon Kil Kwon, Hye-Young Kim, Hyunjung Jeon

**Affiliations:** 1Department of Internal Medicine, Graduate School of Medicine, Chungbuk National University, Chungbuk 28644, Korea; myocarditis@hanmail.net (J.H.C.); kseungjung82@naver.com (S.J.K.); 2Department of Internal Medicine, Chungbuk National University Hospital, College of Medicine, Chungbuk National University, Chungbuk 28644, Korea; hyekim@chungbuk.ac.kr (H.-Y.K.); endoann@daum.net (H.J.)

**Keywords:** GLP-1, GLP-1 receptor, sepsis, acute kidney injury, chronic kidney disease

## Abstract

Acute kidney injury (AKI) is common in patients with sepsis and causes renal ischemia. Glucagon-like peptide-1 (GLP-1) protects the vascular system and the kidney, and GLP-1 receptor (GLP-1R) is expressed in the kidney. Renal GLP-1R activity is decreased in chronic kidney disease (CKD), but is increased by the inflammatory response; however, the effect of AKI on GLP-1R expression is unknown. We investigated the role of GLP-1 by assessing GLP-1R expression in the renal cortex in animals with AKI-related sepsis, CKD, and CKD-with-sepsis. We generated a model of CKD by 5/6 nephrectomy, and sepsis induced by cecal perforation, in male Sprague–Dawley rats. We compared renal GLP-1R expression at 3, 6, 12, 24, and 72 h after cecal perforation, and in CKD and CKD-with-sepsis. We performed blood and urine tests, western blotting (WB), and immunohistochemistry (IHC) to assay GLP-1R expression in renal tubules. The CKD-with-sepsis group showed the lowest kidney function, urine volume, and serum glucose and albumin levels. GLP-1R expression in renal tubules was decreased at 3 h, increased at 24 h, and decreased at 72 h after sepsis induction. GLP-1R expression was decreased at 8 weeks after CKD and was lowest in the CKD-with-sepsis group. The WB results were verified against those obtained by IHC. GLP-1R expression in renal tubules is increased in early sepsis, which may explain the protective effect of endogenous GLP-1 against sepsis-related inflammation.

## 1. Introduction

Sepsis is an important cause of acute kidney injury (AKI), which has a mortality rate approaching 70% [1]. Systemic and intra-renal vasoconstriction causes renal tubular ischemia in sepsis [2], and adequate fluid supply is needed to improve renal perfusion [3]. Although the mechanism of sepsis-related renal damage has been investigated [4,5], it remains unclear. Reliable biomarkers of AKI are also needed.

Glucagon-like peptide-1 (GLP-1) is an incretin hormone produced in the small intestine, which enhances insulin secretion after blood glucose elevation. GLP-1 analogues are used clinically for glycemic control [6]. Moreover, GLP-1 protects vascular endothelial and myocardial cells [7], and exerts a renoprotective effect [8,9]. GLP-1 receptor (GLP-1R) is expressed in the pancreas, central nervous system, heart, and intestine and regulates insulin secretion and vasodilation [10,11]. GLP-1R is also expressed in the glomeruli and proximal tubules of the kidney [12].

The levels of several biomarkers, including GLP-1, are increased in septic patients [13]. Renal tubular epithelial cells contribute to the inflammatory response in ischemic kidney injury by producing inflammatory cytokines [14]. We have reported that GLP-1R expression is enhanced by inhibition of GLP-1 degradation via dipeptidyl peptidase-4 (DPP-4) inhibition, and that GLP-1R activity is decreased in chronic kidney disease (CKD) [15]. However, the role of GLP-1R in the human kidney is unclear. Therefore, we investigated GLP-1R expression in renal tubules in sepsis-related AKI and in CKD.

## 2. Materials and Methods

### 2.1. Animals and Disease Models

Male Sprague–Dawley rats (*n* = 22, 200–250 g body weight in 8 weeks; Daehan Biolink, Chungbuk, South Korea) were used in this study. The required sample size was calculated by Mead’s resource equation; we included several additional animals to cover any incidental perioperative mortality. In the first experiment, the expression of GLP-1R in renal tubules was assessed at 3, 6, 12, 24, and 72 h after sepsis induction. In the second experiment, the rats were divided into control, CKD, sepsis, and CKD-with-sepsis groups. To induce CKD, we removed two-thirds of the left kidney 1 week before the experiment and removed the right kidney at the start of the experiment. To induce CKD-with-sepsis, we triggered artificial septicemia at 8 weeks after induction of CKD. After 3 days, blood and urine, and renal and small-intestinal tissue were collected (Figure 1).

### 2.2. CKD Model

For general anesthesia, 50 mg/kg tiletamine plus zolazepam (Zoletil) and 10 mg/kg xylazine (Rompun) were mixed and injected into the thigh muscle. Next, the surgical area was shaved and disinfected. In all CKD groups, the lower and upper thirds of the left kidneys were resected and after 1 week the right kidney was removed (5/6 nephrectomy); a sham operation was performed in the control group.

### 2.3. Sepsis Model

Under anesthesia as above, after making an intramuscular, fascial, and peritoneal incision, the cecum was located and exteriorized. The total length of the cecum was measured from the tip of the ascending cecum to the tip of the descending cecum. The cecum was ligated at 70% of its total length and perforated by a single puncture midway between the ligation and the tip of the cecum using a 20-G needle. After removing the needle, a small amount of feces was extruded. The cecum was relocated, after which the fascia, abdominal musculature, and peritoneum were closed via simple running sutures; the skin was also sutured. Immediately post-procedure, 1 mL of saline was administered subcutaneously for fluid resuscitation (5 mL/100 g) [16]. Three days after surgery, samples and tissues were collected.

After completion of the experiments, the animals were euthanized by ether inhalation, without pain or stress, based on the standard operating procedure of the Institutional Animal Care and Use Committee (IACUC); the IACUC approved the study protocol (CBNUA).

### 2.4. Animal Survival

The survival rate was analyzed by generating Kaplan–Meier curves using SPSS software (ver. 24; SPSS Inc., Chicago, IL, USA).

### 2.5. Blood and Urine Tests

Urine was collected, and the urine volume over a 24-h period was measured. Creatinine clearance was assessed using a mineral oil-treated metabolic cage on the day of organ harvest. Blood samples were obtained via femoral venous sampling, centrifuged, and stored at −80 °C Body weight and the serum glucose, creatinine, and albumin levels were measured using a Nova Stat Profile M Critical Care Analyzer (Nova Biomedical, Waltham, MA, USA). The pH of fresh urine was measured using an Orion 3 Star Plus pH meter (Thermo, Waltham, MA, USA).

### 2.6. Western Blotting

Renal and intestinal GLP-1R expression was measured in renal cortex and intestinal tissue by Western blotting (WB) using a specific antibody (Bioss, Boston, MA, USA); β-actin (Sigma-Aldrich, St. Louis, MO. USA) was used as the loading control. Proteins were extracted using Pro-Prep protein-extraction solution (Intron, Seoul, South Korea) and assayed spectrophotometrically. Samples were loaded on 10% polyacrylamide-sodium dodecyl sulfate mini gels and transferred to polyvinylidene fluoride membranes. The membranes were blocked for 2 h in Tris-buffered saline, 0.1% plus Tween 20 (TBS-T) containing 5% non-fat dry milk, and treated with primary antibodies against GLP-1R and β-actin for 2 h in TBS-T followed by the secondary goat anti-rabbit horseradish peroxidase-IgG (Santa Cruz Biotechnology, Santa Cruz, CA). WB band densities were quantified using Multi Gauge v. 3.1 software (Fujifilm, Tokyo, Japan) and expressed as percentages relative to the control.

### 2.7. Immunohistochemistry

The kidney and intestine were harvested and fixed in 8% periodate–lysine–paraformaldehyde (PLP) solution for 8 h at room temperature, stored at 4 °C overnight, and embedded in paraffin. For immunohistochemistry (IHC), tissue sections were rinsed in xylene to remove paraffin, and rehydrated in a gradient of 100% to 70% ethanol. Endogenous peroxidase activity was inhibited by treatment with 3% H_2_O_2_ at 4 °C for 45 min. Slides treated with normal goat serum (Vector Laboratories, Inc., Burlingame, CA, USA) were exposed to the primary antibodies at 4 °C overnight, followed by a biotinylated goat anti-rabbit IgG (MACH2 Rabbit HRP Polymer; BioCare Medical, Concord, CA, USA) at room temperature for 30 min. Next, cardiac tissue sections were treated with 3,3′-diaminobenzidine substrate, rinsed with xylene, and mounted. The paraffin blocks were sectioned at 4 µm and stained.

### 2.8. Statistical Analysis

Data are presented as means ± standard deviation. The normality of variables was evaluated using the Shapiro–Wilk Test. The independent sample *t*-test was used to compare the experimental and control groups.

## 3. Results

### 3.1. Clinical and Laboratory Parameters

The sepsis, CKD, and CKD-with-sepsis groups showed significant body weight loss and a lower serum albumin level than the control group (*p* < 0.01); the CKD-with-sepsis group showed the lowest body weight and serum albumin level (Table 1). Creatinine clearance in the sepsis group was significantly decreased compared to the control, and that in the CKD group was lower than in the sepsis group; creatinine clearance was lowest in the CKD-with-sepsis group (*p* < 0.01). Although the plasma C-reactive protein level was increased in the CKD and sepsis groups, the increase was significant only in the sepsis group (*p* = 0.068 vs. *p* = 0.024). The 24-h urine volume was increased in the CKD group but markedly decreased in the CKD-with-sepsis group (Figure 2).

The 70-day-survival rate of the control, sepsis, CKD, and CKD-with-sepsis groups was 100% (*n* = 4), 66.7% (*n* = 6), 100% (*n* = 5), and 71.4% (*n* = 7), respectively (Figure 3).

### 3.2. GLP-1R Expression in Renal Tubules Was Increased in Early Sepsis

GLP-1R expression in the renal cortex was low at 3 h after cecal perforation and subsequently increased to a peak at 24 h. GLP-1R expression in the renal cortex decreased to lower than the control at 3 days after sepsis (Figure 4B). IHC showed that GLP-1R expression in renal tubules decreased at 3 h (Figure 4C, b), increased at 24 h (Figure 4C, e), and decreased at 72 h (Figure 4C, f).

### 3.3. Renal GLP-1R Expression Was Decreased in Sepsis and CKD-with-Sepsis

GLP-1R expression in the renal cortex was significantly decreased in the sepsis and CKD-with-sepsis groups compared to the control. The GLP-1R band density was non-significantly decreased in the CKD group compared to the control (Figure 5B). The sepsis-with-CKD group showed decreased GLP-1R expression in the renal cortex compared to the CKD-only group. IHC showed that renal GLP-1R expression was decreased in the sepsis (Figure 5C, b), CKD (Figure 5C, c), and CKD-with-sepsis (Figure 5C, d) groups. The CKD-with-sepsis group showed the lowest tubular GLP-1R activity (Figure 5C, d).

### 3.4. Intestinal GLP-1R Expression Was Markedly Decreased in the Sepsis, CKD, and CKD-with-Sepsis Groups

The band density of GLP-1R in intestinal tissue was markedly decreased in the sepsis, CKD, and CKD-with-sepsis groups compared to the control. Intestinal GLP-1R activity is thought to be lower than that in the renal cortex in patients with sepsis and CKD-with-sepsis. However, there were no significant differences among the groups (Figure 6).

## 4. Discussion

GLP-1R expression in renal tubules was increased in early sepsis and later decreased by kidney injury, but was decreased in the CKD and CKD-with-sepsis groups. This is, to our knowledge, the first study of changes in renal GLP-1R expression in sepsis. GLP-1 is an incretin hormone that shows increased expression after food intake; it protects the vascular system and myocardium by increasing myocardial insulin sensitivity [7]. GLP-1 agonists have beneficial effects on cardiovascular mortality and ameliorate kidney injury in diabetics [17], possibly by protecting the endothelium. Also, GLP-1 secretion is increased in the presence of acute inflammation [18] and in critically ill diabetic patients [13], which might be an adaptive physiologic response to stress. GLP-1 improves vascular dysfunction in sepsis [19], but Perl et al. found that diabetic patients with sepsis who showed an extremely early increase in the GLP-1 level had a poor prognosis, suggesting that severe acidemia activates endogenous GLP-1 [13].

GLP-1R is a class-B G-protein coupled receptor (GPCR) with a large extracellular domain [20]; its signaling is mediated by stimulation of the G-protein pathway [21]. Kulve et al. reported that decreased GLP-1R expression in the hypothalamus is associated with dysregulation of glucose control in diabetics [22]. GLP-1R is also expressed in the kidney glomeruli and tubules [8], and suppression of GLP-1 degradation suggested that it has a renoprotective role [23,24]. Renal GLP-1R expression was shown to be increased by DPP-4 in experimental animals [25], as we also reported in an earlier study [15]. Furthermore, DPP-4 inhibition reduces proteinuria, ameliorates renal-function impairment [26], and decreases apoptosis [27]. However, the mechanism underlying the decrease in proteinuria mediated by GLP-1 and GLP-1R in the kidney is unclear. We found that renal GLP-1R activity was increased in early sepsis, possibly due to acute inflammation, which may decrease after tissue injury. However, the decrease in renal GLP-1R activity at 3 days might not be directly associated with tubular injury because the CKD group, which had a higher serum creatinine level, exhibited higher renal GLP-1R expression than the sepsis-only group (Figure 5B).

Renal GLP-1 exerts a renoprotective effect by enhancing tubular sodium excretion via blocking sodium–hydrogen exchanger (NH3) [28] and decreasing angiotensin activity in healthy volunteers [29]. Elsewhere, a GLP-1 agonist decreased renal sodium reabsorption in type 2 diabetes without influencing the renal hemodynamics [30]. Furthermore, Huang et al. reported that GLP-1 pathways inhibit the extracellular matrix expression of mesangial cells [31], and Skov et al. demonstrated that GLP-1 downregulates the tissue angiotensin II system [32]. Although we could not demonstrate changes in renal blood flow or sodium excretion in sepsis, the increase in GLP-1R expression in early sepsis might explain the early increase in the angiotensin II level [33]. Renal tubular and myocardial GLP-1R expression was increased by DPP-4 inhibition in CKD and acute myocardial infarction [15], supporting a correlation between inflammation and altered GLP-1R expression. Further investigation of the changes in GLP-1 and GLP-1R expression in the kidney is warranted.

This study had several limitations. First, the small number of experimental animals may have resulted in certain changes being overlooked. However, we evaluated sepsis at 3, 6, 12, 24, and 72 h, and compared the changes in GLP-1R expression between CKD and CKD-with-sepsis groups (and where the results were consistent among the timepoints). Second, we could not compare the serum GLP-1 and DPP-4 levels between the sepsis and CKD groups because plasma GLP-1 is rapidly degraded by DPP-4, explaining why the change in serum GLP-1 level in sepsis cases based on the blood GLP-1 concentration is problematic. We compared changes in GLP-1R expression in the kidney among animals with sepsis, CKD, and CKD-with-sepsis; however, further studies of DPP-4 in sepsis are warranted. Third, we could not evaluate the interaction between GLP-1 and GLP-1R. Therefore, more research on related molecular markers and signaling pathways is needed to clarify the role of GLP-1 and GLP-1R in the kidney.

## 5. Conclusions

In conclusion, GLP-1R expression is increased in early sepsis, which may explain renal endogenous GLP-1 activation, and is decreased at the late stage of sepsis and in CKD. Induction of GLP-1R expression may protect against inflammation and sepsis-induced AKI.

## Figures and Tables

**Figure 1 ijms-20-06024-f001:**
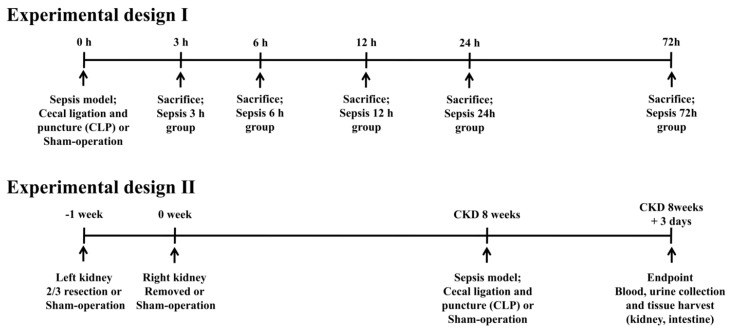
Experimental design. Control, sham-operated; sepsis, cecal ligation, and puncture (CLP) model; and CKD, 5/6 nephrectomized model.

**Figure 2 ijms-20-06024-f002:**
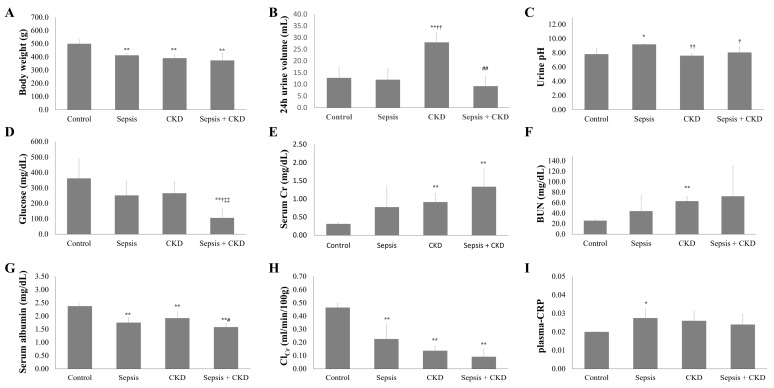
Changes of clinical parameters in rats. Change of outcomes in control, sepsis, CKD, and CKD-with-sepsis rats. Body weight (**A**), 24 h urine volume (**B**), urine pH (**C**), glucose (**D**), serum creatinine (**E**), blood urea nitrogen (**F**), serum albumin (**G**), creatinine clearance (**H**), and plasma-CRP (**I**). Control, sham-operated; sepsis, cecal ligation and puncture (CLP) model; and CKD, 5/6 nephrectomized model. *n* = number of rats. * *p* < 0.05 vs. control, ** *p* < 0.01 vs. control; † *p* < 0.05 vs. sepsis, †† *p* < 0.01 vs. sepsis; and # *p* < 0.05 vs. CKD, ## *p* < 0.01 vs. CKD.

**Figure 3 ijms-20-06024-f003:**
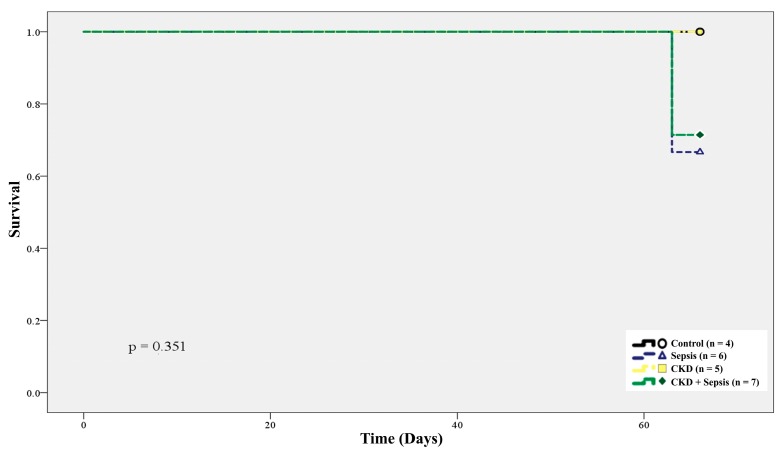
Kaplan–Meier plot of conditions of control, sepsis, CKD, and CKD-with-sepsis associated with animal survival rate.

**Figure 4 ijms-20-06024-f004:**
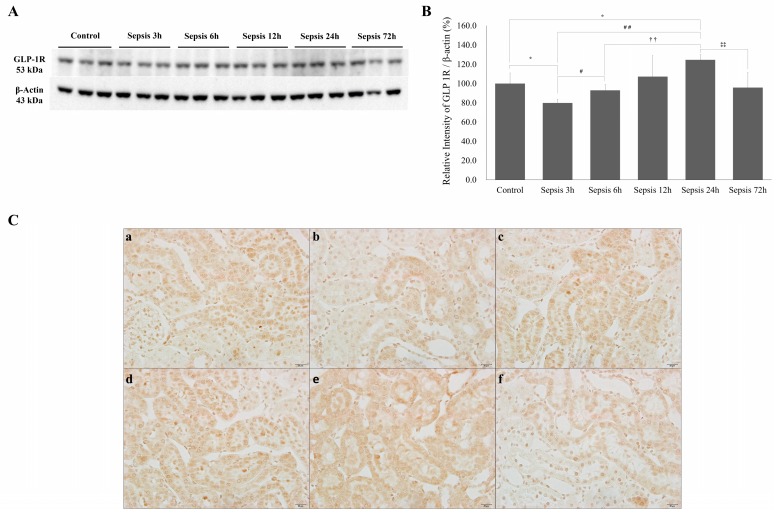
Renal GLP-1R changes in early sepsis. (**A**) Western blots showing GLP-1R expression in the renal cortex. (**B**) Quantification of GLP-1R was standardized based on β-actin expression in the renal cortex. (**C**) Immunohistochemistry of renal GLP-1R (×400). (**a**) Control, (**b**) 3 h after cecal perforation, (**c**) 6 h, (**d**) 12 h, (**e**) 24 h, and (**f**) 72 h after cecal perforation. Control, sham-operated; sepsis, cecal ligation, and punctured (CLP) rats; values are presented as the mean ± standard deviation (SD). * *p* < 0.05 vs. control, # *p* < 0.05 vs. sepsis 3 h, ## *p* < 0.01 vs. sepsis 3 h, †† *p* < 0.01 vs. sepsis 6 h, and ‡‡ *p* < 0.01 vs. sepsis 24 h.

**Figure 5 ijms-20-06024-f005:**
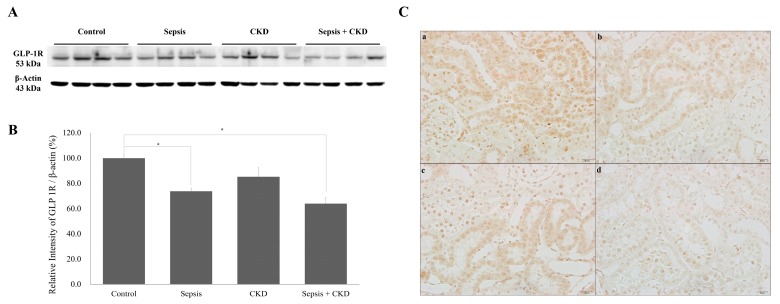
Renal GLP-1R expression in sepsis, CKD, and CKD-with-sepsis. (**A**) Western blots showing GLP-1R expression in the renal cortex. (**B**) Quantification of GLP-1R was standardized based on β-actin expression in the renal cortex. (**C**) Immunohistochemistry of renal GLP-1R (×400). Control, sham-operated; sepsis, cecal ligation, and punctured (CLP) rats; values are presented as the mean ± standard deviation (SD). * *p* < 0.05 vs. control.

**Figure 6 ijms-20-06024-f006:**
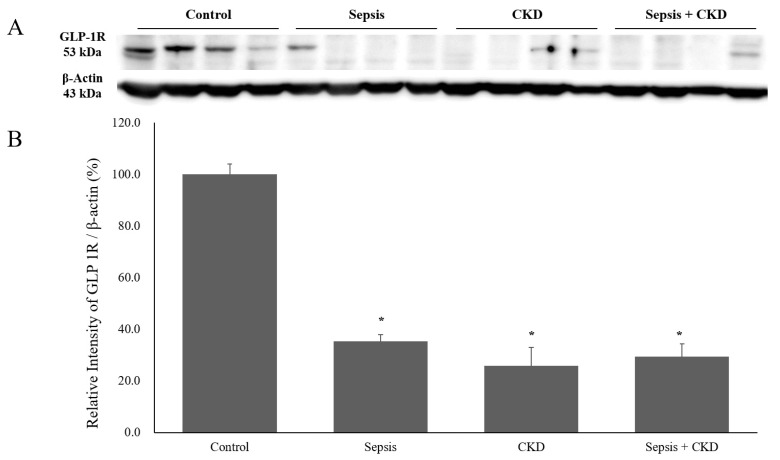
Intestinal GLP-1R in sepsis, CKD, and CKD-with-sepsis. (**A**) Western blot showing intestinal GLP-1R expression. (**B**) Quantification of GLP-1R was standardized based on β-actin expression in the intestine. Sepsis, cecal ligation, and punctured (CLP) rats; values are presented as the mean ± standard deviation (SD). * *p* < 0.05 vs. control.

**Table 1 ijms-20-06024-t001:** Clinical parameters of control, sepsis, chronic kidney disease (CKD), and CKD-with-sepsis rats.

Variables	Control (*n* = 4)	Sepsis (*n* = 6)	CKD (*n* = 5)	Sepsis + CKD (*n* = 7)
Body weight (g)	500.0 ± 29.7	412.5 ± 10.6 ^**^	390.6 ± 24.7 ^**^	373.6 ± 55.5 ^**^
24 h urine volume (mL)	12.8 ± 2.5	12.0 ± 4.3	28.0 ± 4.9 ^**,††^	9.2 ± 4.3 ^##^
Urine pH	7.83 ± 0.87	9.2 ± 0.08 ^*^	7.60 ± 0.28 ^††^	8.07 ± 0.84 ^†^
glucose (mg/dL)	362.8 ± 126.2	251.7 ± 92.3	265.8 ± 73.8	106.0 ± 61.9 ^**,†,##^
Serum Cr (mg/dL)	0.31 ± 0.03	0.78 ± 0.59	0.91 ± 0.23 ^**^	1.34 ± 0.50 ^**^
BUN (mg/dL)	25.9 ± 2.5	44.0 ± 29.2	63.0 ± 9.4 ^**^	72.4 ± 57.7
Serum albumin	2.4 ± 0.1	1.8 ± 0.2 ^**^	1.9 ± 0.2 ^**^	1.6 ± 0.1 ^**,#^
Cl_Cr_ (ml/min/100 g)	0.46 ± 0.03	0.23 ± 0.12 ^**^	0.14 ± 0.04 ^**^	0.09 ± 0.06 ^**^
Plasma-CRP	0.02 ± 0.00	0.03 ± 0.01 ^*^	0.03 ± 0.01	0.02 ± 0.01

Cr, creatinine; BUN, blood urea nitrogen; ClCr, creatinine clearance; plasma-CRP, plasma-C-reactive protein; control, sham-operated; sepsis, cecal ligation and puncture (CLP) model; CKD, 5/6 nephrectomized model; values are presented as the mean ± standard deviation (SD). *n* = number of rats. * *p* < 0.05 vs. control, ** *p* < 0.01 vs. control; † *p* < 0.05 vs. sepsis, †† *p* < 0.01 vs. sepsis; and # *p* < 0.05 vs. CKD, ## *p* < 0.01 vs. CKD.

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
