# Peer review of "Renal Tubular Glucagon-Like Peptide-1 Receptor Expression Is Increased in Early Sepsis but Reduced in Chronic Kidney Disease and Sepsis-Induced Kidney Injury"

_ijms, 2019, doi:10.3390/ijms20236024_

Round 1

Reviewer 1 Report

The major contribution of this paper is considering the renal tubular Glucagon-like peptide-1 receptor expression in early sepsis and sepsis-induced kidney injury. 

I recommend that this paper be accepted after minor revision.

Minor comments:

Line 54: How many weeks old rats were used in the experiment?

Table 1:

I think the blood sugar in the control group of SD rats is too high.

I think the CRP of Sepsis group and Sepsis + CKD group are too low.

Are these true?

Figure 5:

I think figure legend is typo. (About statistics.)

Figure 6:

How about the amount of intestinal GLP-1R in early sepsis?

Author Response

[Reviewer 1]

The major contribution of this paper is considering the renal tubular Glucagon-like peptide-1 receptor expression in early sepsis and sepsis-induced kidney injury. 

I recommend that this paper be accepted after minor revision.

Minor comments:

Line 54: How many weeks old rats were used in the experiment?

The rats were about 8 weeks of 200 ~ 250 g weight. We aded the age of the animal on the manuscript.

Table 1:

I think the blood sugar in the control group of SD rats is too high.

The blood sugar level was post-prandial. However, we agree with your opinion as usual post-prandial rat glucose is around 100 ~ 150 mg/dL (6 mmol/L). We think this is strange finding, but we have similar sugar level in our previous experiment cited in this manuscript [Kim SJ et al, BMC Nephrology 2019 20:75] which might be associated with experimental condition (environment) or animal-animal difference. However, CKD rats showed decreased blood sugar in both experiment (BMC Nephrology article and this manuscript).

I think the CRP of Sepsis group and Sepsis + CKD group are too low.

Are these true?

CRP level was also not too higher than expected in sepsis only model. So we think too low level of CRP in CKD with sepsis was also nonspecific finding. We just report as it originally found. Sorry about confusing data, but it was factual result.

Figure 5:

I think figure legend is typo. (About statistics.)

It was a big mistake. We apologize the error, and we corrected it. Thank you.

Figure 6:

How about the amount of intestinal GLP-1R in early sepsis?

We did not plan to investigate earlier intestinal GLP-1R changes in this experiment, so we could not compare earlier changes. Also, intestinal GLP-1R difference comparing with control group were much higher than those of the kidney.

Reviewer 2 Report

This clear and well design study and well written manuscript describes the kinetic of GLP-1 receptor expression in kidney (and gut) of CKD rats with and without sepsis.

Few comments to the authors :

-the number of animals in each group should be given in table 1

-In figure 2 authors should specify on the axial axis "survival" and number of days on the longitudinal axis.

-a statistical point to clarify : the authors compared the mean of the groups in each experiment by T-test. Have they proven that the distribution of the different variables followed a Gaussian distribution ? if not they should instead use a non parametric Mann and Whitney test

Author Response

[Reviewer 2]

This clear and well design study and well written manuscript describes the kinetic of GLP-1 receptor expression in kidney (and gut) of CKD rats with and without sepsis.

Few comments to the authors :

-the number of animals in each group should be given in table 1

We added the number of the animals on table 1. Thank you.

-In figure 2 authors should specify on the axial axis "survival" and number of days on the longitudinal axis.

We think you pointed figure 3. We added explanation of X and Y axis. Thank you.

-a statistical point to clarify : the authors compared the mean of the groups in each experiment by T-test. Have they proven that the distribution of the different variables followed a Gaussian distribution ? if not they should instead use a non parametric Mann and Whitney test

Our description was insufficient and not clear. We previously checked the data for checking normality using Shapiro-Wilk Test, and found those are consistent with normality (data in table).

variable

Body weight

Urine Vol

Urine pH

Glucose

S-Cr

BUN

ALB

plasma-CRP

U-Cr

Cl-Cr

control

0.996205

0.979792

0.7851

0.941222

0.537997

0.799887

0.905211

0.000670925

0.950868

0.932027

sepsis

0.934798

0.90625

0.937811

0.99539

0.828545

0.716813

0.999685

0.416735045

0.995486

0.99405

CKD

0.97086

0.759098

0.961855

0.99966

0.91616

0.996085

0.310342

0.509556965

0.851111

0.961516

CKD with Sepsis

0.701206

0.991827

NA

0.454865

0.902453

0.804654

0.921912

0.509556965

0.849511

0.915896

After your good comment, we added below on statistical method section.

The normality of variables was evaluated using the Shapiro-Wilk Test.